# Research Trends in the Efficacy of Stem Cell Therapy for Hepatic Diseases Based on MicroRNA Profiling

**DOI:** 10.3390/ijms22010239

**Published:** 2020-12-29

**Authors:** Minyeoung Kweon, Jae Yeon Kim, Ji Hye Jun, Gi Jin Kim

**Affiliations:** 1Department of Molecular and Cellular Biology, University of Glasgow, Glasgow G12 8QQ, UK; minyeoungkweon@gmail.com; 2Department of Biomedical Science, CHA University, Seongnam 13488, Korea; janejykim92@gmail.com (J.Y.K.); jihyejun1015@gmail.com (J.H.J.)

**Keywords:** degenerative diseases, liver failure, microRNAs, stem cell therapy

## Abstract

Liver diseases, despite the organ’s high regenerative capacity, are caused by several environmental factors and persistent injuries. Their optimal treatment is a liver transplantation. However, this option is limited by donor shortages and immune response issues. Therefore, many researchers have been interested in identifying the therapeutic potential in treating irreversible liver damage based on stem cells and developing suitable therapeutic agents. Mesenchymal stem cells (MSCs), which are representative multipotent stem cells, are known to be highly potential stem cell therapy compared to other stem cells in the clinical trial worldwide. MSCs have therapeutic potentials for several hepatic diseases such as anti-fibrosis, proliferation of hepatocytes injured, anti-inflammation, autophagic mechanism, and inactivation of hepatic stellate cells. There are much data regarding clinical treatments, however, the data for examining the efficacy of stem cell treatment and the correlation between the stem cell engraftment and the efficacy in liver diseases is limited due to the lack of monitoring system for treatment effectiveness. Therefore, this paper introduces the characteristics of microRNAs (miRNAs) and liver disease-specific miRNA profiles, and the possibility of a biomarker that miRNA can monitor stem cell treatment efficacy by comparing miRNAs changed in liver diseases following stem cell treatment. Additionally, we also discuss the miRNA profiling in liver diseases when treated with stem cell therapy and suggest the candidate miRNAs that can be used as a biomarker that can monitor treatment efficacy in liver diseases based on MSCs therapy.

## 1. Introduction

The liver is an important organ in that it is responsible for various physiological processes such as glucose, lipid, and cholesterol metabolism as well as immune system support [1,2]. The hepatectomized or injured liver can grow back to its original size through the proliferation of the epithelium and the stroma due to their high regenerative capacity [3]. When the liver undergoes partial hepatectomy, more than a hundred genes are activated and prepare hepatocytes to respond to growth factors. Liver damage is also caused by drugs, viruses, and toxins. These lead to chronic liver pathologies like non-alcoholic fatty liver disease and cirrhosis. The damaged liver stimulates the activation of hepatic progenitor cells (HPCs) which are thought to contribute to the regeneration of the damaged liver tissue [4]. In the early stage of liver diseases, inflammation starts since the immune system is trying to fight infection and heal an injury. If the inflammation is consistent, the liver undergoes fibrosis where scar tissue accumulates. As the hepatic stellate cells activate, they transform into myofibroblast-like cells and synthesize extracellular matrix proteins. The extracellular matrix abnormally accumulates which progresses liver fibrosis. This leads to liver failure and making the liver to overburden to heal the scarred part. If untreated, the liver progresses to cirrhosis which is irreversible [5,6] (Figure 1).

Chronic liver diseases, including liver cirrhosis, are known to display high morbidity and mortality [7]. To date, liver damage is usually monitored by measuring the level of alanine transaminase (ALT), aspartate aminotransferase (AST), alkaline phosphatase (ALP), albumin, and bilirubin using blood chemistry [8]. So far, liver transplantation is the gold standard protocol when the liver undergoes irreversible damage, however, organ shortages remain a challenge for liver transplantation [9]. Thus, there is a need to regenerate and replace damaged tissues using new strategic stem cell-based treatments. Extracorporeal liver support devices have been applied to support the native liver to recover itself from the injury especially patients with acute liver failure [10] and acute-on-chronic liver failure (ACLF). There are two kinds of devices that are potentially useful for treatment to patients with liver failure. Artificial liver devices use artificial components to replace detoxification functions by using different kinds of membranes. However further studies are required to prove the benefits in clinical uses and detoxification alone is not enough. Another one is bio artificial liver devices. Bio-artificial liver systems provide both biotransformation and synthetic liver functions by using hepatocytes in a bioactive form. However, limitations such as loss of cell viability and functionality and cost have delayed its use [11,12].

Hepatogenic differentiated cells derived from stem cells or hepatocytes isolated from liver tissues can be used to enhance desired biological liver functions in injured liver tissues. However, the limitations are that the cells are harvested in low quantity and have low proliferation potential [13]. Therefore, stem cell therapy has shown critical insights into treatment, especially in end-stage liver diseases. Several kinds of stem cells, mesenchymal stem cells (MSCs) are used commonly since they lack ethical issues [14].

Generally, stem cells exist endogenous in our body that have clonogenic potential to self-renew and differentiate into multiple cell lineages. They can be divided according to their source and their differential potency. They mainly come from embryonic tissue, fetal tissues, amniotic fluid, umbilical cord, placenta and genetically reprogrammed somatic cells [15]. In addition, compared to other somatic cells that do not replicate themselves, they can proliferate. Thus, this characteristic is useful in laboratories since an abundant amount of stem cells can be produced just from a little amount of stem cells [16]. Due to their unique characteristics, they are thought to be one of the revolutionary discoveries in medicine. Especially, cancer stem therapy has received attention because cancer treatment was only limited to drugs, surgery, antibiotics, and radiation. However, the stem cells have the capacity to transport medicine to the damaged tissue [17].

To add, stem cells can target the damaged tissue in degenerative pathologies. MSCs have the potential to treat neurodegenerative diseases such as Alzheimer’s, Parkinson’s, and Huntington’s disease as well as are known to take over or support damaged neurons by strengthening the immune system and secreting neurotrophic factors [18]. A lot of diseases and disorders are still being studied and plenty of answers are left to be answered. With the help of stem cells, solutions are uncovered. Research based on stem cells continues to develop and identify potential therapeutic factors. One of the diseases that are still being studied in this context is hepatic diseases [19]. Previous studies have found that MSCs act as an effective therapy for chronic liver diseases and a low dosage of MSCs can enhance liver function. MSCs can repress inflammation, decrease hepatocyte apoptosis, increase hepatocyte regeneration, and slow down liver fibrosis [20].

MicroRNAs (miRNAs) are noncoding small-sized RNA molecules (21–25 nucleotides) that target mRNA and regulate protein expression. The secreted miRNAs maintain stability in body fluids and can be isolated for examination for related diseases. MiRNAs also display diagnostic tools for degenerative diseases such as neurodegenerative disorders and liver failure and are receiving attention as important biomarkers [21,22]. As several degenerative diseases including Alzheimer’s disease (AD), circulating miRNA levels increase which is considered as potential non-invasive biomarkers for diagnosis and prognosis. Studies have shown that miRNAs react with related proteins such as amyloid-β (Aβ) and genes like tau, Rb1, and other genes [23,24].

Thus, this review introduces the characteristics of miRNAs and liver disease-specific microRNA profiles. Additionally, we also introduce miRNA profiling in liver diseases when treated with stem cell therapy and suggest candidate miRNAs that can be used as biomarkers to monitor treatment efficacy in liver diseases based on MSC therapy.

## 2. Main body

### 2.1. Classifications of Stem Cells

The discovery of stem cells is initiated by the migration of cells in the mouse bone marrow to the spleen and the formation of individual colonies when transplanted to a heavily irradiated mouse recipient. This colony as colony-forming units-spleen (CFU-S) had the variable capacity to self-regenerate and possessing heterogeneity within the colony [25]. Stem cells can renew themselves and differentiate into different cell types during development. They differ in their degree of plasticity or developmental versatility [26].

Stem cells can be categorized by their potential to generate multiple types of specialized cells; totipotent, pluripotent, multipotent, and unipotent (Figure 2). Totipotent stem cells are formed after the fertilization of an egg cell by a sperm cell. They are the most versatile stem cell type being able to differentiate into cells of the whole organism. A zygote is a totipotent cell that forms later into three germ layers or a placenta. After a few days, totipotent cells specialize into pluripotent stem cells. Pluripotent stem cells (PSCs) can regenerate by dividing and develop into the three primary germ cell layers of the early embryo and thus into all cells of the adult body. The limitation of PSCs is the unavailability to produce extraembryonic tissues like the placenta [27,28].

Embryonic stem cells (ESCs) and induced PSCs (IPSCs) are pluripotent stem cells. ESCs can differentiate into all the tissues of the embryo and adult. Human embryonic stem cell lines (hESCs) are from the inner cell mass of blastocysts. This process requires the damage to the embryo and thus has given rise to ethical problem [29]. Thus, IPSCs are known to have overcome these issues. IPSCs are reprogrammed back into an embryonic-like pluripotent state from fully differentiated cells such as fibroblasts and have similar proliferative and developmental characteristics as ESCs. IPSCs come from adult cells like the skin or blood cells and can be reprogrammed back into an embryonic-like pluripotent state through transcription factors like Oct4, cMyc, Sox2, and Klf4. Thus, this has the capacity to develop into any type of human cell [30]. This provides the advantage to differentiate into patient-specific somatic cells and treat specifically [31]. Recent studies have proved that IPSCs can differentiate into hepatocytes and are applied to liver disorders such as non-alcoholic and alcoholic liver diseases, viral hepatitis, and cirrhosis [32].

Adult or somatic stem cells are multipotent stem cells. Multipotent stem cells are more specific that pluripotent stem cells. They can differentiate into many cell types but show a restricted pattern of differentiation toward few lineages. Among several multipotent stem cells, hematopoietic stem cells (HSCs) can form into various types of blood cells but not into the brain or liver cells [33]. Especially, hematopoietic stem cells can differentiate into myeloid stem cells and lymphoid stem cells. After that, myeloid stem cells can differentiate into erythrocytes, platelets, and myeloblasts. Lymphoid stem cells differentiate into lymphoblast and later produce lymphocytes and white blood cells. MSCs as known non-hematopoietic stem cells are also multipotent stem cells that can be isolated from tissues and organs of the human body such as adipocytes, myocytes, and neurons. They have several advantages such as multipotent, immunosuppressive properties, and no ethical problem. The barrier of MSC is the lack of tissue sources and difficult and invasive retrieval methods [34].

### 2.2. Stem Cell Therapy in Degenerative Diseases

Stem cell therapies are being an effective treatment for various degenerative diseases such as Parkinson’s disease, degenerative disc disease and heart failure. The stem cell can prevent disease or decelerate the disease or even cure the disease [35]. MSCs have displayed the capacity to differentiate into osteoblasts, adipocytes, and chondrocytes in specific environments [36]. Also, it has been shown that MSCs display anti-proliferative and anti-inflammatory effects [37]. MSCs can inhibit the potential of pro-inflammatory effects of dendritic cells by decreasing their production of tumor necrosis factor (TNF). In addition, anti-inflammatory cytokine interleukin-10 (IL-10) is upregulated when plasmacytoid dendritic cells are incubated with MSCs. [38].

MSCs also contribute to the therapy in fibrotic diseases in the kidney, lung, heart, liver, skin, and bone marrow [39]. The key risk factors of fibrosis are fibroblasts and myofibroblasts because they are involved in the synthesis of extracellular matrix proteins in injured tissues for fibrotic processing. In the case of the liver, the production of extracellular matrix proteins is mainly mediated by activated HSCs. When the liver is damaged, the quiescent HSCs change into proliferative, α-smooth muscle actin (α-SMA)-positive, myofibroblast-like cells with increased collagen production. They can down-regulate myofibroblasts and lead to anti-fibrotic activity. Thus, many clinical trials of MSCs as an anti-fibrotic treatment in liver cirrhosis are in progress [40]. In addition, MSCs are able to inhibit the proliferation of activated HSCs and collagen synthesis by secreting trophic factors such as IL-10, hepatocyte growth factor (HGF), transforming growth factor beta 3 (TGF-β3), and TNF-α. In addition, MSCs can be co-cultured with HSCs which suppresses the proliferation and expression of α-SMA [41,42].

### 2.3. Liver Diseases

The liver is a critical organ for metabolism, excretion and the immune system in the body. It plays a key role in biochemical reactions such as storage, degradation, and synthesis. When the liver is not working properly, the whole body is affected [43,44]. Especially, genetic liver diseases are the result of the accumulation of various substances from an abnormal gene passed down from one or both parents. Liver diseases classified as a metabolic disease include Wilson’s disease, hemochromatosis, and α-1 antitrypsin deficiency [45]. Hepatitis is a viral infection of the liver and one of the main causes of ALF and chronic liver disease. There are five types of hepatitis according to their transmission routes and health disorders, which are hepatitis A to E. Hepatitis A, B are known to account for three-quarters of all ALF cases. These usually cause inflammation and scarring the liver tissue [46,47]. Also, abnormal hepatic lipid metabolism advances to fatty liver diseases such as non-alcoholic fatty liver disease (NAFLD) and alcoholic fatty liver disease (AFLD). Hepatic lipid metabolism is regulated by various factors like hormones, transcription factors and inflammatory cytokines [48]. Simple steatosis accounts for the majority of NAFLD cases and non-alcoholic steatohepatitis (NASH) accounts for the rest of them. Compared to simple steatosis, NASH can advance to cirrhosis and hepatocellular carcinoma (HCC). NAFLD patients are considered as metabolic disease patients and normally have diabetes showing insulin resistance [49]. Advanced liver and cirrhosis are key causes of high mortality and these lead to HCC and decompensated cirrhosis. The major risk factors of HCC are hepatitis B or C virus infection or alcohol abuse [50].

As shown above, there are many different types of liver diseases but they all have a common progress. Once the liver is damaged, the liver has difficulty functioning in a normal way. In the early stage of any liver disease, inflammation is caused since the immune system is working to fight or heal an infection. If the inflamed liver is neglected, it progresses to fibrosis. The key cause for the outset and progression of liver fibrosis is oxidative stress and the resulting inflammatory responses [51]. The liver cells keep repairing the frequent and continuing injury that ends up leaving scar tissue. The liver stiffens because HSCs are activated and transformed into myofibroblast-like cells to synthesize extracellular matrix proteins such as collagens. Excessive amount of extracellular matrix and collagen accumulates. The fibrous material distribution is based on what liver injury is caused. For example, the fibrotic tissue is first found around the portal tracts in chronic viral hepatitis and chronic cholestatic disorders. In alcohol-induced liver disease, it is in the pericentral and perisinusoidal areas. The blood supply is unable to reach to the tissue and progresses to cirrhosis [51,52,53,54]. The scar tissue starts to enlarge and builds up in the organ. Liver cirrhosis is caused by the severe scarring of the liver and an advanced state of liver fibrosis. The liver architecture is distorted and increased portal blood flow results in portal hypertension. Hepatic cirrhosis is the irreversible end stage disease of liver and transplantation is the main solution. Finally, chronic liver cirrhosis leads to HCC [6,55].

Liver disease treatments are focused on inhibiting the activation of stellate cells and inflammation and removing the cause of the harm. Steroids are used to restrict the liver tissue damage and decrease the molecular signals for fibrosis. They also help to degrade the extracellular matrix [56]. Liver transplantation is a therapy where a diseased liver is replaced with a healthy liver. It is recommended when the liver can no longer function properly and is at the end-stage of chronic liver disease. However, the challenges of liver transplantation are the shortage of organs and the rejection response of transplanted organs [57].

### 2.4. Stem Cell Therapy for Liver Diseases

Due to the easy accessibility and the known safety of MSCs, they are widely used in clinical trials. Studies have found enhanced mesenchymal stem cells have the power to improve liver fibrosis, reverse hepatic cirrhosis and liver diseases and enhance liver function. MSCs can differentiate into hepatocytes and produce anti-inflammatory responses and modulate the immune responses. In addition, MSCs can stimulate liver regeneration by elevating the hepatocyte proliferation and reduce pro-inflammatory and fibrogenic cytokine activity at the same time [58]. In the histopathological appearance, MSC-treated livers show decreased hepatocellular death and liver shape distortion. MSCs can reduce the activation of hepatic stellate cells and have anti-inflammatory effects. Also, MSCs secrete anti-fibrosis factor such as matrix metalloproteinase-2 MMP-2 and the angiogenic factor including vascular endothelial growth factor (VEGF). In addition, MSCs can make the macrophages proliferate into an anti-inflammatory phenotype by secreting factors such as prostaglandin E2 and IL-13. These produce metalloproteinases that lyses extracellular matrix and phagocytosis increases which can phagocytize the hepatocyte debris. Data showed that MSC therapy significantly improves liver function in patients with liver disease when analyzing the model of end-stage liver disease score, albumin, alanine aminotransferase, and total bilirubin levels, and prothrombin time [59,60,61]. Secretomes derived from MSCs are classified by various growth factors and cytokines such as TGF-β3, HGF, and IL-10, exosomes and miRNAs. These factors can attenuate liver fibrosis. The secretome also includes extracellular vesicles (EV) that have therapeutically beneficial effects. The EV contains lipids, proteins, DNA, and RNA molecules. They are able to express MSC surface markers that can change the immune response [62,63].

Clinical studies have proved that MSC transplantation in liver diseases results in improvement in liver functions. Stem cells are mainly transplanted via peripheral veins. They can also be transplanted through intrasplenic, intrahepatic, and portal vein. Xu et al. examined the potency of umbilical cord-derived mesenchymal stem cell combined with plasma exchange treatment in hepatitis B virus-related acute on chronic liver failure patients. They identified improvement in ALT, AST, and bilirubin levels and suggested the capacity of such technique in long term treatment to patients. [64,65]. Another study also examined the effect of MSC in auto immune disease induced liver cirrhosis patients. The Models of End Stage Liver Disease (MELD) scores were improved and the study emphasized the safety of MSC in clinical trials. Similar study found the efficacy of bone marrow-derived mesenchymal stem cell transplantation to alcoholic liver cirrhosis patients. They transplanted the stem cell to the hepatic artery and examined improvement in histologic fibrosis [64,66,67].

### 2.5. MicroRNA

MiRNAs are RNA molecules that are approximately 19 to 28 nucleotides (nt) in size and participate in regulating gene expression. MiRNAs are important in that they are related to a variety of biological processes such as development, cell proliferation, and differentiation [68]. The miRNA gene is transcribed by RNA polymerases II or III to pri-miRNA with a cap and a poly-A tail. Then, it is processed into short 70-nt stem-loop structures known pre-miRNAs by a protein complex. Pre-miRNAs are processed to miRNA duplexes by the RNase III enzyme Dicer. The miRNA duplex is accompanied by the protein Argonaut (AGO) and forms an RNA-induced silencing complex (RISC). This complex target mRNA to regulate gene expression by translational repression or mRNA degradation [69]. AGO-2 and GW182 proteins are needed to repress mRNAs in the early stage by interacting negatively with translation initiation machinery which includes eIF6. Processing blood are subcellular foci where some of the mRNAs are located in and stored or decapped and degraded [70].

MiRNAs are secreted out of the cell by exosomes and microvesicles. These are found in different types of body fluid such as blood, serum/plasma, cerebrospinal fluid, synovial fluid, urine, and saliva. In addition, miRNA expression is different depending on the cells, tissues and organs involved. Some miRNAs are specific or abundant in certain organ tissues and thus specify diseases such as cancer. To add, miRNA expression act as biomarkers for physiological and pathological changes. Since circulating miRNAs are tissue and organ specific and disease and stage specific, they act as noninvasive biomarkers [71] (Figure 3).

### 2.6. Monitoring Stem Cell Therapy

To understand and examine the role and fate of stem cell therapy, monitoring the location of the stem cells and their biological interaction with the surrounding is important. Thus, studies have demonstrated various techniques for monitoring implanted stem cells and analyzing cell migration, differentiation and redistribution. Stem cell engraftment is an important process for the stem cells to survive after transplantation. This is a process where stem cells arrive at the bone marrow to proliferate and produce red blood cells, white blood cells and platelets. Stem cell engraftment can be enhanced by naturally derived or synthetic polymeric biomaterials [72,73]. To examine stem cell engraftment, tracking down the stem cells is essential to determine the success of the stem cell to the target. Various techniques like nanoparticles and exosomes are applied for implanted stem cell monitoring but have limitations like toxicity and not being biodegradable. MiRNAs are known to participate in effective engraftment by regulating angiogenesis and HSC after stem cell transplantation and thus, miRNA expression changes significantly [74].

Nanoparticles (NP) can be used for stem cell trafficking and imaging and are effective when used with imaging techniques such as tomography, ultrasound or photoacoustic imaging. NP technology can track stem cells for a long time and displays high sensitivity in tracking stem cells [75]. One study used ultrasound-guided photoacoustic imaging to image gold nanotracer-labeled mesenchymal stem cells and found that the labeled MSCs can be detected with high sensitivity and over a long period [76]. Gold nanoparticles come in various sizes and surface coatings and are effective nanotracers because of their good biocompatibility. MSCs can be coated with gold nanoparticles and traced for a long time without any alteration of cell function [77,78]. However, nanoparticles are not biodegradable and thus future studies must consider the possibility of chronic toxicity and the long-term side effects. In addition, evidence has been found that NPs can affect stem cells by activating or inhibiting differentiation [79].

Exosomes are also applied as non-invasive biomarkers. Exosomes are 40–100 nm nanosized vesicles that are widely distributed in various body fluids and secreted from many cell types such as immune cells, stem cell, and cardiovascular cells. Lipids and proteins are usually contained in the vesicles and various nucleic acids such as mRNAs, miRNAs, and non-coding RNAs are also components. Exosomes play an important role in cell-to-cell communication by interacting between cells and affect the physiological pathways in the recipient cells [80,81]. Therefore, exosomes are used as potential biomarkers since the measurement of protein and RNA contents are measured differently compared with normal and patients. For example, miR-192 levels in serum exosomes were upregulated in cardiovascular disease patients and can be applied as a prediction for heart failure [82]. Moreover, exosomes are related to inflammatory processes that are involved in diseases like cancer, diabetes, obesity, liver disease, and neurodegenerative disease and thus the expression level of some exosomes is crucial biomarkers for these inflammatory based diseases [83]. Furthermore, studies have found that stem cell-induced exosomes can serve as biomarkers and treatments for diseases like cardiovascular, neurological diseases, and cancer. Previous studies have noted that stem cells secrete exosomes that interact with the tumor and the tumor microenvironment and influmonuence the pathway of the tumorigenesis, tumor angiogenesis, and tumor metastases. The use of mesenchymal stem cells to AD patients has received attention and MSC-derived exosomes are effective therapeutic methods. However, increasing evidence has found that exosomes can spread toxic factors like Aβ, and hyperphosphorylated tau and sometimes leading to cell apoptosis [84,85].

MiRNAs are related to various cellular activities such as cell proliferation and apoptosis. Especially, miRNAs are also correlated to self-renewal of stem cell and stem cell differentiation and thus are useful for monitoring stem cell differentiation in the absence of intoxicating the cells [86]. One study found that by examining the miRNA expression, MSC that express high levels of neurotrophic factors can be distinguished from their original MSCs in amyotrophic lateral sclerosis (ALS) patients and thus can be used as criteria and show future implications for ALS studies. Upon the analyzed miRNAs, miR-132-3p was upregulated and this miRNA is known to be correlated with neuronal differentiation and axonal extension. In addition, miR-320a, miR-424-5p, and miR-503 were down-regulated and these miRNAs are associated with VEGF signaling [87]. Another study compared the expression of miRNAs of bone marrow mesenchymal stem cell-derived micro vesicles between young and old rats and identified that miR-133b-3p and miR-294 are related to renal epithelial-to-mesenchymal transition and aging [88]. Therefore, such studies indicate that miRNA profiling shows the role of stem cells and the pathway of the target genes.

There is a lack of a monitoring system for how liver diseases recover with stem cell therapy. In addition, due to molecular and genetic differences between people, they may respond differently to treatment even though they have the same disease and thus, biomarkers need to be used in clinical applications [89]. MiRNAs act as specific biomarkers in liver diseases since they display dramatic and fast changes compared to traditional biomarkers such as ALT and AST [90].

### 2.7. MicroRNA in Degenerative Diseases

Exosomes have high specific and stability and thus are known to be useful as potential biomarkers. MiRNAs are enriched in exosomes and widely secreted in the peripheral body fluids. Data from several studies suggest that miRNAs display abnormal the level patterns in various diseases such as cancer, degenerative diseases and neurodegenerative diseases. This suggests that circulating miRNAs can be used as a monitoring system for certain diseases [91,92,93]. Xu et al. identified that the downregulation expression of miR-27a-3p in ALS patients indicating that miRNAs are related with the development of ALS and applied as early diagnostic markers [94]. In addition, Lu et al. showed that mice with degenerative joint diseases displayed increased miR-23a/b-3p expression and Grem1 downregulation. Thus, they targeted these factors by inhibiting the miR-23a/b-3p or overexpressing the Grem1 which resulted in attenuating the progression of the degenerative joint diseases [95]. Divi et al. compared the circulating miRNA levels between patients with degenerative disc disease and healthy controls and found that miR-155-5p acted as a diagnostic marker because it was significantly downregulated and displayed high correlation with the pro-inflammatory cytokine IL-1β and the tumor suppressor genes p53 and BRAF [96]. Ma et al. found that downregulation of miR-181c-5p and miR-497-5p levels indicated postmenopausal women with osteopenia or osteoporosis. Researchers validated this fact by using aging and ovariectomized mice models and suggested that overexpression of the miRNAs promote the differentiation and mineralization of osteoblasts [97]. AD is also one of the neurodegenerative diseases and still needs further finding for drugs to prevent the progression of the disease and thus early diagnostic is essential. MiRNAs are enriched in the brain and is important in biological functions in neurons such as neuron differentiation and neurogenesis. MiRNAs like miR-26b-3p, miR-28–3p, and miR-30c-5p are upregulated and let-7a-5p, miR-15a-5p, miR-1294, and miR-3200–3p are downregulated [98]. Shi et al. reported that expression of miRNA shows better understanding of the mechanisms of the degenerative aortic stenosis disease. They found that six miRNAs (hsa-miR-193a-3p, has-miR-29b-1-5p, hsa-miR-505-5p, hsa-miR-194-5p, hsa-miR99b-3p, and hsa-miR-200b-3p) were upregulated and 14 miRNAs (hsa-miR-3663-3p, hsa-miR-513a-5p, hsa-miR-146b-5p, hsa-miR-1972, hsa-miR-718, hsa-miR-3138, hsa-miR-21-5p, hsa-miR-630, hsa-miR-575, hsa-miR-301a-3p, hsa-miR-636, hsa-miR-34a-3p, hsa-miR-21-3p, and hsa-miR-516a-5p) were downregulated in patients with AS compared to normal controls [99].Therefore, miRNAs are potential biomarkers to monitor the progression of diseases and understand the various targets and factors of the diseases. MiRNAs can be detected by real-time PCR, microarrays and sequencing technologies and are mostly found in body fluids such as the blood, cerebrospinal fluid, and urine.

### 2.8. MicroRNAs in Liver Disease

Table 1 presents an overview of miRNAs that play critical roles in normal liver development and liver regeneration. MiRNAs are involved in regulating multiple factors during liver development such as hepatocyte proliferation and differentiation and metabolism. MiRNAs regulate multiple factors of liver development and regeneration. One miRNA can be involved in multiple fields such as miR-10a is involved in definitive endoderm formation and liver detoxification.

In addition, miR-122 has a role in hepatocyte proliferation and differentiation, liver glucose metabolism and liver lipid metabolism. MiRNAs are also related to liver regeneration. After liver transplantation or acute liver injury, the liver is able to regenerate itself through a complex process and miRNAs are known to regulate the pathways involved and promote or repress liver regeneration (Table 1).

As mentioned, miRNAs play a role as biomarkers in degenerative diseases and among them are liver disease such as hepatitis, NAFLD, fibrosis, cirrhosis and HCC. MiRNA are biomarkers that inform about the damage to the liver and further can differentiate between the different types of hepatic disease (Table 2) [100]. Among the miRNAs, miR-122 accounts for 70 percent of the total hepatic miRNA. It is also known to be expressed in hepatic diseases such as HBV and HCV infection and the levels of the expression are associated with the performance of the disease. Thus, studies have been ongoing to target miR-122 for therapeutic reasons [101]. Additionally, for HCV patients, miR-122, miR-134, miR-424-3p, and miR-629-5p were upregulated [102]. Furthermore, chronic hepatitis B (CHB) [103] patients also expressed altered levels in miRNA compared to healthy controls. Among them, miR-122, miR-572, miR-575, miR-638, and miR-74 were nominated to be potential biomarkers for CHB patients. Besides, ALT levels were neutral in CHB patients, but miRNAs were differently displayed and therefore proposing that miRNAs are promising biomarkers in hepatitis [104,105]. Likewise, NAFLD patients are also can be diagnosed by looking at miRNA levels since miRNA is thought to act in lipid metabolism and homeostasis [106]. Serum levels of miR-122, miR-34a, and miR-16 are upregulated in NAFLD patients when compared to healthy controls. NASH is the progressive form of NAFLD and miR-122 and miR-34a levels show the degree of severity of NASH [107]. NASH can be also discriminated from simple steatosis by the down-regulated expression of miR-122 [108].

Liver fibrosis is the result of the redundant accumulation of extracellular matrix produced by HSCs following inflammation. Liver biopsy and several biomarkers such as ALT or AST are the references to diagnose liver fibrosis and cirrhosis [109]. With the emergence of miRNA, the strong relationship between miRNAs and liver fibrosis has been observed and thus miRNAs have become diagnostic and prognostic biomarkers. For example, the miR-29 family (miR-29a, miR-29b, and miR-29c) showed down-regulated expression in a CCl_4_-induced liver fibrosis murine model. On the other hand, miR-34 members (miR-34a, miR-34b, and miR-34c) were found to be up-regulated [110]. Early diagnosis of liver fibrosis is also important for the sake of inhibiting the progression to liver cirrhosis. MiR-122 and miR-200b have been suggested to be biomarkers for the early diagnosis of liver fibrosis [111].

miRNAs can also show the efficacy of treatment when stem cells are used to treat the liver. The miRNAs are differentially expressed in different types of liver diseases [112]. When stem cells are treated, the level of miRNAs changes compared to when injured. MiR-122 levels indicate the efficacy of bone marrow-derived stem cells. Hepatic miR-122 levels were upregulated in the CCl_4_-induced model. After stem cell therapy, miR-122 levels showed a decrease [113]. Also, miR-21 was shown to be upregulated in many types of MSCs-derived stem cells and thus is an effective therapeutic target.

One study found that miR-30e expression was downregulated in CCl_4_-induced hepatic fibrosis model and identified that HuR regulates the S1P induced bone marrow-derived MSCs motility and suggested a new regulatory mechanism for liver fibrogenesis [114].

**Table 1 ijms-22-00239-t001:** miRNAs involved in liver development and liver regeneration.

**I. Liver Development**		
**Function**	**Target miRNAs**	**Reference**
Definitive endoderm formation	miR-10a, miR-24, miR-196a, miR-196b, miR-218, miR-222, miR-338-5p, miR-340-3p, miR-371-3p, miR-371-5p, miR-373, miR-375, miR-520	[115,116,117,118,119]
Epithelial- to-mesenchymal transition (EMT)	miR-30b/c/d/e, miR-218, miR-495	[120,121,122,123,124,125]
Hepatocyte proliferation and differentiation	miR-33, miR-122, miR-148a	[126,127,128,129,130]
HSC activation and proliferation	miR-16, miR-19a, miR-27a, miR-27b, miR-29b, miR-146a, miR-150, miR-194, miR-195, miR-335	[110,131,132,133,134,135,136,137,138,139,140]
Liver glucose metabolism	miR-27b, miR-33, miR-96, miR-122, miR185, miR-206, miR-223, miR-370	[141,142,143,144,145,146,147,148]
Liver iron metabolism	miR-485-3p	[149,150]
Detoxification	miR-10a, miR-21, miR-24, miR-27b, dre-miR-27b, miR-34a, miR-93, miR-126, miR-130, miR-132, miR-142-3p, miR-148a, miR-185, miR-200c, miR-214, miR-378, miR-699, miR-892a	[151,152,153,154,155,156,157,158,159]
**II. Liver Regeneration**		
**Function**	**Target miRNAs**	**Reference**
Regeneration promoters	miR-19a, miR-21, miR-214, miR-106a, miR-20a, miR-20b, miR-93, miR-33, miR-153, miR-298, miR-301b, miR-489, miR-743b, miR-883, miR-126, miR-130a, miR-20a, miR520e	[160]
Regeneration inhibitors	let-7a/b/c/d/e/f/i, miR-23a/b, miR-26a/b, miR-29a, miR-30d, miR-33, miR-125b-5p, miR-126, miR-127, miR-145, miR-146a, miR-150, miR-207, miR-223, miR-352, miR-375, miR-378, miR-503, miR-532-3p, miR-663, miR-872	[160]

**Table 2 ijms-22-00239-t002:** Alterations in miRNA expression in liver diseases.

Disease	Target miRNAs	Reference
Chronic Hepatitis C (CHC)	miR-21, miR-23b, miR-27a, miR-106, miR-122, miR-134, miR-320c, miR-424-3p, miR-451, miR-629-5p, miR-638, miR-762, miR1207-5p, miR-1225-5p, miR1275, miR-1246 miR-1974	[102,161,162]
Chronic Hepatitis B	miR-16, miR-19b, miR-20a, miR-92a, miR-106a, miR-122, miR-122-3p, miR-125b, miR-146a-5p, miR-194, miR-223, miR-328-3p, miR-572, miR-575, miR-638, miR-744	[163,164,165,166]
NAFLD	miR-16, miR-21, miR-27b-3p, miR-34a, miR-122, miR-122-5p, miR-145, miR-192-5p, miR-451, miR-1290	[167,168,169]
Liver fibrosis	miR-29, miR-29a, miR-34a, miR-199a, miR-200a, miR-200b, miR-571, miR-513-3p, miR-652	[110,170,171]
Liver cirrhosis	miR-29, miR-106b, miR-181b, miR-513-3p, miR-571, miR-652, miR-885-5p	[100,172,173]
Hepatocellular Carcinoma (HCC)	miR-15b, miR-18a, miR-21, miR-26a-5p, miR-29b, miR-106b, miR-122, miR-122-5p, miR-122a, miR-130b, miR-141-3p, miR-143, miR-183, miR-192-5p, miR-193a-3p, miR-199a-5p, miR-206, miR-215, miR-369-5p, miR-429, miR-433-5p, miR-483-5p, miR-672, miR-1228-5p	[174,175,176]

Overlapping miRNAs: miR-10a, miR-122, miR-33, miR-148a, miR-27b, miR-16, miR-29b, miR-194, miR-34 a/b/c, miR-126, miR-143, miR-223, miR-206, miR-21, miR-130b, miR-185, miR-34 a/b/c, miR-638, miR-451, miR-106 a/b, miR-199a, miR-134, miR-424-3p, miR-6395p, miR-192-5p, miR-572, miR-575, miR-638, miR-74, miR-29 a/b/c.

Our previous study analyzed miRNA profiles on bile duct ligation (BDL)-induced cirrhotic liver tissues post-MSC transplantation at 1 week and 2 weeks and migrated MSCs under hypoxic conditions. The expression of miRNAs (miR-7-5p, miR-199a-5p, miR-148a-5p, and let-7c-5p) was related to liver regeneration by phosphatase of regenerating liver-1 (PRL-1) functionally enhanced placenta-derived mesenchymal stem cells (PD-MSCs^PRL-1^) (Figure 4A,B). Especially, fibrosis-related gene expressions (TGF-β1, zinc finger E-box binding homeobox 2; ZEB2, desmoplakin, and E-cadherin) were analyzed with each targeting miRNAs (miR-25-3p, miR-200a-3p, miR-30a-5p, and miR-22-3p) (Figure 5A–D). In histopathological analysis through Sirius red staining, the nontransplantation (NTx) group indicated collagen accumulation. According to PD-MSC and PD-MSC^PRL-1^ transplantation, it was decreased. Interestingly, the PD-MSC^PRL-1^ group displayed significantly decreased Sirius red positive areas compare to the PD-MSC group (Figure 5E,F; *p* < 0.05). The miRNA functioned as a therapeutic target and the therapeutic mechanisms were based on miRNA-mediated stem-cell therapy in liver diseases [115].

## 3. Conclusions

Stem cells mainly come from adult body tissues or embryos and can differentiate into specific cells. They can be organized according to their plasticity, totipotency, pluripotency, multipotency and unipotency and as the stages proceed, they become specific, so they can act as any type of cell and repair damaged tissues. The capabilities of stem cells are still growing, and more studies must be performed. Stem cell therapy has been investigated in several degenerative disorders and mesenchymal stem cells in particular are applied since they secrete effective factors. The liver is a regenerative organ and up to a certain injury level, the organ is able to recover itself and factors like antifibrotic and anti-inflammatory therapy are the key solutions at these reversible stages. However, when the liver undergoes liver cirrhosis, liver injury inhibits the regeneration and damages the liver which is an irreversible stage. Stem cells can differentiate into hepatocytes and produce factors that stimulate repair and regeneration. MSCs have been proven to reverse hepatitis, cirrhosis and liver damage effects by regressing the activation of hepatic stellate cells and secreting anti-inflammatory factors. In addition, anti-fibrotic and angiogenic factors are secreted by the MSCs that help slow down the progression of liver fibrosis and cirrhosis.

Tracking stem cells is important to identify the biological function of stem cells and evaluate their maximized therapeutic efficacy on the target tissue. There are diverse monitoring techniques such as nanoparticles, exosomes, and miRNAs. Among them, miRNAs are the most sensitive and high specific and have high agility compared to other methods and thus represent a promising marker for tracking down stem cells. To diagnose liver diseases, traditional biomarkers such as ALT and AST have been used but they present limitations such as insufficient sensitivity and specificity for diagnosis. As an alternative, miRNAs are emerging biomarkers for such diseases. According to a lot of studies, miRNAs have been demonstrated as an attractive tool. Circulating miRNAs play a crucial role in diagnosing liver diseases and thus have the potential to be non-invasive markers. Basically, several miRNAs participate in liver development and liver regeneration and their expression changes in hepatic diseases. Thus, miRNAs profiling can also act as a therapeutic target as well as the treatment of stem cells in liver disease can be examined by using miRNAs. This paper summarized the miRNAs related to liver development, regeneration and liver diseases. Among the miRNAs found, several miRNAs overlap in some categories such as miR-122, miR-33, and miR-27b. As tracking down stem cells is possible by miRNAs and diseases can be diagnosed through miRNAs, monitoring the stem cell efficacy at the target tissue is possible. This review therefore emphasizes that miRNAs play a crucial role in monitoring the therapeutic effects of stem cells and proving its efficacy and success at the target tissue which leads to the suggestion that miRNAs can indicate the effectiveness and symptoms after stem cell transplantation

## Figures and Tables

**Figure 1 ijms-22-00239-f001:**
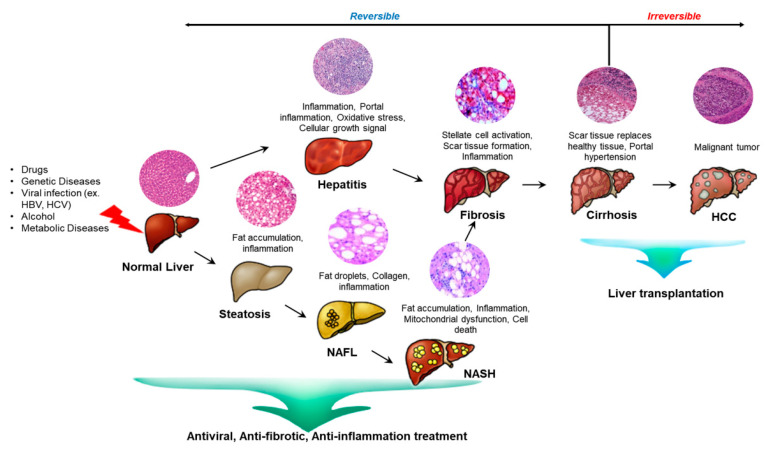
Progressive stages of liver diseases. Liver diseases that lead to HCC are characterized by steatosis and viral hepatitis which is coincided with inflammation and hepatocellular death. Until the liver undergoes fibrosis, reversible wound healing such as antiviral, anti-fibrotic, and anti-inflammation therapy is feasible. However, if the damage carries on, liver diseases may advance to end-stage liver diseases such as cirrhosis and HCC.

**Figure 2 ijms-22-00239-f002:**
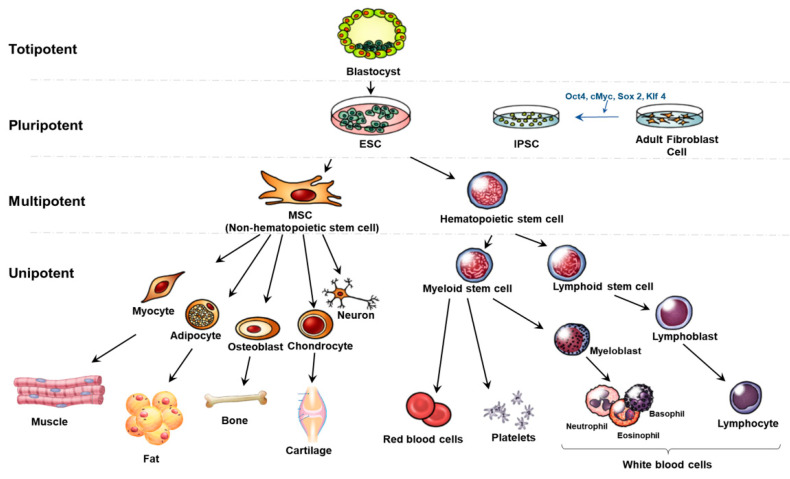
Derivation and differentiation of stem cells. ESCs are usually derived from the inner cell of blastocyst and IPSCs are produced by reprogramming adult cells by factors such as Oct4 and Sox2. Both ESCs and IPSCs are pluripotent and can differentiate in a wide range of specialized cell types which are multipotent stem cells. The multipotent stem cells give rise to specific cell types which include adipocytes, neurons, blood cells and the immune system.

**Figure 3 ijms-22-00239-f003:**
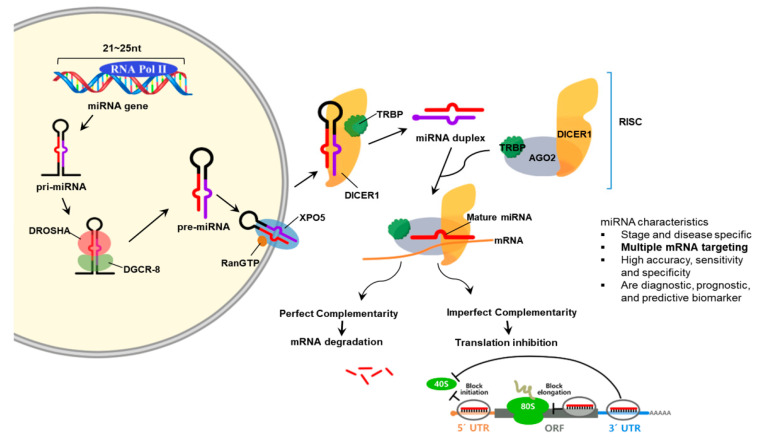
Schematic illustration of miRNA biogenesis. In the nucleus, the miRNA gene is transcribed and the primary miRNA is produced. The primary miRNA is process by the endonuclease, Drosha diver with its cofactor, DGCR8. The stem-looped structure, pre-miRNA, is transported out of the nucleus by exportin and further processed to mature miRNA by the dicer enzyme. The RISC is produced and matches with the mRNA through the complementary sequences and further processes to mRNA degradation or translation inhibition according to the complementarity with the mRNA.

**Figure 4 ijms-22-00239-f004:**
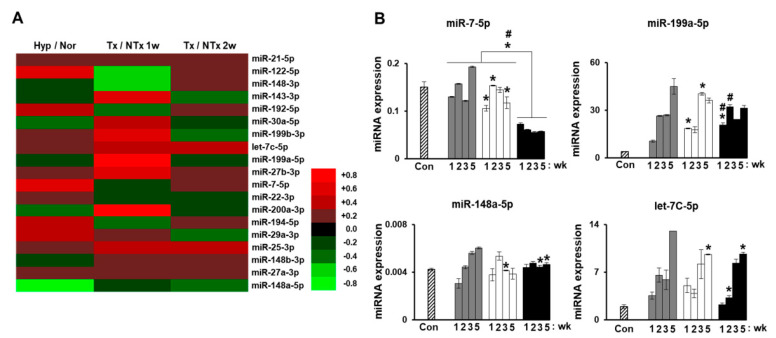
MiRNA validation of migrated placenta-derived mesenchymal stem cells (PD-MSCs) under hypoxic conditions and in bile duct ligation (BDL)-injured liver in rats. (**A**) Heat map of microarray results in migrated PD-MSCs under hypoxic versus normoxic conditions (Hyp/Nor), Tx versus NTx group at 1 week (Tx/NTx 1w), and Tx versus NTx group at 2 weeks (Tx/NTx 2w). (**B**) rno-miR-7-5p, 199a-5p, 148a-5p, and let-7C-5p expression in rat liver with BDL at 1, 2, 3, and 5 weeks PD-MSCs post transplantation by qRT-PCR. Data from each group are expressed as means ± SD. Statistical analysis was performed using Student’s t test and *p* values less than 0.05 were considered statistically significant. *, *p* < 0.05, vs. NTx; #, *p* < 0.05, vs. PD-MSCs; PD-MSCs, naïve PD-MSCs transplanted group; PD-MSCs^PRL-1^, PRL-1 functionally enhanced PD-MSCs transplanted group; wk, week.

**Figure 5 ijms-22-00239-f005:**
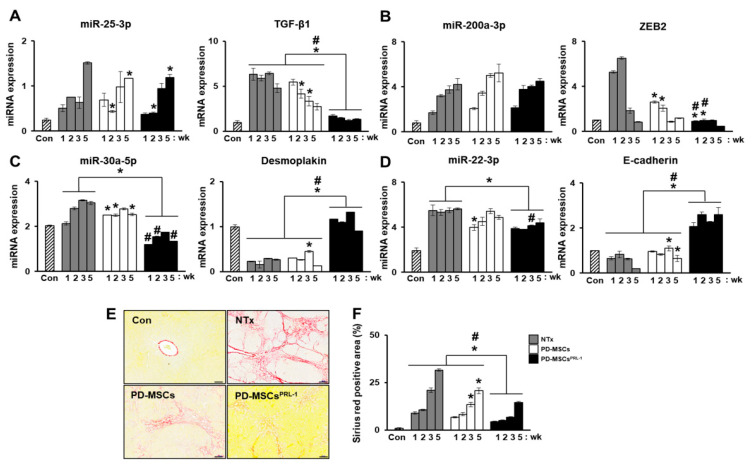
MiRNAs regulates hepatic fibrosis via their target genes in a rat model of BDL. (**A**) TGF-β1-targeted rno-miR-25-3p, (**B**) zinc finger E-box binding homeobox 2 (ZEB2)-targeted rno-miR-200a-3p, (**C**) desmoplakin-targeted rno-miR-30a-5p, (**D**) E-cadherin-targeted rno-miR-22-3p expression in rat liver with BDL at 1, 2, 3, and 5 weeks PD-MSCs post-transplantation by qRT-PCR. (**E**) Sirius Red staining in BDL-injured rat liver tissues from each group (Con, NTx, PD-MSCs, and PD-MSCs^PRL-1^) at 3 weeks. (**F**) Quantification of the accumulated collagen by Sirius Red staining. Scale bars = 200 μm. All experiments were conducted at least triplicate. Data from each group are expressed as means ± SD. Statistical analysis was performed using Student’s t test and *p* values less than 0.05 were considered statistically significant. *, *p* < 0.05, vs. NTx; #, *p* < 0.05, vs. PD-MSCs; PD-MSCs, naïve PD-MSCs transplanted group; PD-MSCs^PRL-1^, PRL-1 functionally enhanced PD-MSCs transplanted group; wk, week.

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
