# Peer review of "Research Trends in the Efficacy of Stem Cell Therapy for Hepatic Diseases Based on MicroRNA Profiling"

_ijms, 2020, doi:10.3390/ijms22010239_

Round 1

Reviewer 1 Report

In this review article, the authors introduced the characteristics of microRNAs (miRNAs) and liver disease-specific miRNA profiles, and the possibility of a biomarker that miRNA can monitor stem cell treatment efficacy by comparing microRNAs changed in liver diseases following stem cell treatment. The authors also discussed the miRNA profiling in liver diseases when treated with stem cell therapy and suggest the candidate miRNAs that can be used as a biomarker that can monitor treatment efficacy in liver diseases based on MSCs therapy. Overall, the manuscript could have interesting implications. Specific points need to be considered are listed below:

  1. The authors need to include more previous findings about stem cell therapy /microRNA profiling in clinical studies.
  2. It’s unclear whether the exact same data in Figure 4 and 5 has been previously published by the authors. If not, the authors need to provide the Methods and Materials for Figure 4 and 5.
  3. The introduction of microRNA should be included or before paragraph 2.5.
  4. It’s better to expand paragraph 2.8 to different kinds of liver diseases.
  5. The title of paragraph 2.3 should be ‘Liver Diseases’. The authors also need to check the spelling for ‘disease/diseases’ in the text.

Author Response

Manuscript ID: 1033272

Reviewer #1

General comments: In this review article, the authors introduced the characteristics of microRNAs (miRNAs) and liver disease-specific miRNA profiles, and the possibility of a biomarker that miRNA can monitor stem cell treatment efficacy by comparing microRNAs changed in liver diseases following stem cell treatment. The authors also discussed the miRNA profiling in liver diseases when treated with stem cell therapy and suggest the candidate miRNAs that can be used as a biomarker that can monitor treatment efficacy in liver diseases based on MSCs therapy. Overall, the manuscript could have interesting implications. Specific points need to be considered are listed below:

  • Author’s reply: We greatly appreciate the reviewer’s positive statement that "Overall, the manuscript could have interesting implications".

Point #1: The authors need to include more previous findings about stem cell therapy /microRNA profiling in clinical studies.

  • Author’s reply (1): We thank you for the reviewer’s comment and we agree that further data in section 2.4 is needed and thus added to the manuscript which is denoted in red.

Point #2: It’s unclear whether the exact same data in Figure 4 and 5 has been previously published by the authors. If not, the authors need to provide the Methods and Materials for Figure 4 and 5.

  • Author’s reply (2): Thank you for the reviewer’s comments. We analyzed the same samples in our previous reports.
  • [Kim et al., Dynamic Regulation of miRNA Expression by Functionally Enhanced Placental Mesenchymal Stem Cells Promotes Hepatic Regeneration in a Rat Model with Bile Duct Ligation. Int J Mol Sci. 2019. 20(21):5299.]

Point #3: The introduction of microRNA should be included or before paragraph 2.5.

  • Author’s reply (3): We thank you for pointing out the rearrangement. Section 2.6 which is the introduction of microRNA has been replaced before section 2.5. Corrected parts are marked in red.

Point #4: It’s better to expand paragraph 2.8 to different kinds of liver diseases.

  • Author’s reply (4): We thank you for the reviewer’s comment. To address that point, please refer to table 2 in the manuscript.

Point #5: The title of paragraph 2.3 should be ‘Liver Diseases’. The authors also need to check the spelling for ‘disease/diseases’ in the text.

  • Author’s reply (5): We appreciate the reviewers comment. We corrected it and added it in the revised manuscript. Corrected parts are marked in red.

Reviewer 2 Report

In this review the Kweon et al. discuss the relevance of microRNAs as biomarkers that can monitor stem cell treatment efficacy in liver diseases. I consider this manuscript suitable for IJMS audience. However, there are some issues that should be addressed. They are as follows:

Major

Many unnecessary abbreviations are used once or twice that make reading difficult; e.g. HPCs, ALP, ACLF, Aβ, CFU-S, hESCs, MMP-2, VEGF, EV, AD, NTFs, AGO, HBV, HCV. Conversely, there are important ones not included in the abbreviation section; e.g. HCC hepatocellular carcinoma, HSCs hematopoietic stem cells.

The authors provide excessive information regarding the use of microRNAs as biomarkers in non-hepatic diseases. I suggest removing the text from line 345 to 368.

Minor

Line:

  20: …and so on…       avoid using run on expressions

  36: …a hundred of genes…        provide reference/s

  42: …delete or use alternative abbreviation for hepatic stellate cells

  97: …(21-25 nucleotides)…       mentioned again in line 311

  98: …miRNAs are to maintain…        miRNAs maintain

127: ESCs and induced PSCs (iPSCs) are pluripotent stem cells. ES cells…        

        Embrionic stem cells (ESCs) and induced PSCs (IPSCs) are pluripotent stem cells. ESCs

134: …factors like oct4, cMyc, Sox2, and Klf4…       transcription factors like Oct4, c-Myc, Sox2, and Klf4

185: ...this can affect our whole body….

189: …ALF       already defined in line 64

193: …nonalcoholic…       non-alcoholic

272: …40-100nm…       40-100 nm

311: …that are 19 to 25 nucleotides…       that are approximately 19 to 28 nucleotides (nt)

318: …AGO-2 and GW182 are…       Argonaut-2 and GW182 proteins are

464: … ant fibrotic…      anti-fibrotic

Figure 4 and 5: The statistical test used to calculate the p-value should be indicated.

Author Response

Manuscript ID: 1033272

Reviewer #2:

General comments: In this review the Kweon et al. discuss the relevance of microRNAs as biomarkers that can monitor stem cell treatment efficacy in liver diseases. I consider this manuscript suitable for IJMS audience. However, there are some issues that should be addressed. They are as follows:

  • Author’s reply: We greatly appreciate the reviewer’s positive statement that " I consider this manuscript suitable for IJMS audience.".

Point #1: Many unnecessary abbreviations are used once or twice that make reading difficult; e.g. HPCs, ALP, ACLF, Aβ, CFU-S, hESCs, MMP-2, VEGF, EV, AD, NTFs, AGO, HBV, HCV. Conversely, there are important ones not included in the abbreviation section; e.g. HCC hepatocellular carcinoma, HSCs hematopoietic stem cells. The authors provide excessive information regarding the use of microRNAs as biomarkers in non-hepatic diseases. I suggest removing the text from line 345 to 368.

  • Author’s reply (1): We appreciate the reviewer’s points. As you commented, abbreviations have been added in the manuscript. Corrected parts are marked in red.The information regarding the use of microRNAs as biomarkers in degenerative diseases was to inform about the critical role of RNAs as potential biomarkers and to provide general background that miRNAs can be used as targets in various diseases. And among those degenerative diseases, we aimed to focus on liver diseases. Thus, we suggest that this information is needed regardless of excessive information as the reviewer has suggested.

Point #2:

Line:

  20: …and so on…       avoid using run on expressions

  36: …a hundred of genes…        provide reference/s

  42: …delete or use alternative abbreviation for hepatic stellate cells

  97: …(21-25 nucleotides)…       mentioned again in line 311

  98: …miRNAs are to maintain…        miRNAs maintain

127: ESCs and induced PSCs (iPSCs) are pluripotent stem cells. ES cells…       

        Embrionic stem cells (ESCs) and induced PSCs (IPSCs) are pluripotent stem cells. ESCs

134: …factors like oct4, cMyc, Sox2, and Klf4…       transcription factors like Oct4, c-Myc, Sox2, and Klf4

185: ...this can affect our whole body….

189: …ALF       already defined in line 64

193: …nonalcoholic…       non-alcoholic

272: …40-100nm…       40-100 nm

311: …that are 19 to 25 nucleotides…       that are approximately 19 to 28 nucleotides (nt)

318: …AGO-2 and GW182 are…       Argonaut-2 and GW182 proteins are

464: … ant fibrotic…      anti-fibrotic

Figure 4 and 5: The statistical test used to calculate the p-value should be indicated.

  • Author’s reply (2): We appreciate the reviewer’s comments. These are corrected and added in the manuscript. Descriptions about statistical analysis added in figure legend. Corrected parts are marked in red.

Round 2

Reviewer 1 Report

The authors have addressed all previous comments.